# Healthcare utilization in children across the care continuum during the COVID-19 pandemic

**Alan R. Schroeder**[1]*, **Alex Dahlen**[2], **Natasha Purington**[2], **Francisco Alvarez**[1], **Rona Brooks**[1], **Lauren Destino**[1], **Gayatri Madduri**[1], **Marie Wang**[1], **Eric R. Coon**[3]

**1** Department of Pediatrics, Stanford University School of Medicine, Palo Alto, CA, United States of America, **2** Department of Medicine, Quantitative Sciences Unit, Stanford University School of Medicine, Palo Alto, CA, United States of America, **3** Department of Pediatrics, Primary Children's Hospital and University of Utah School of Medicine, Salt Lake City, UT, United States of America

* aschroe@stanford.edu

**Data Availability Statement:** Database is proprietary (owned by Optum) and only available for purchase through the company. The Population Health Sciences Department of the Stanford

## Abstract

### Objectives

Healthcare utilization decreased during the COVID-19 pandemic, likely due to reduced transmission of infections and healthcare avoidance. Though various investigations have described these changing patterns in children, most have analyzed specific care settings. We compared healthcare utilization, prescriptions, and diagnosis patterns in children across the care continuum during the first year of the pandemic with preceding years.

### Study design

Using national claims data, we compared enrollees under 18 years during the pre-pandemic (January 2016 –mid-March 2020) and pandemic (mid-March 2020 through March 2021) periods. The pandemic was further divided into early (mid-March through mid-June 2020) and middle (mid-June 2020 through March 2021) periods. Utilization was compared using interrupted time series.

### Results

The mean number of pediatric enrollees/month was 2,519,755 in the pre-pandemic and 2,428,912 in the pandemic period. Utilization decreased across all settings in the early pandemic, with the greatest decrease (76.9%, 95% confidence interval [CI] 72.6–80.5%) seen for urgent care visits. Only well visits returned to pre-pandemic rates during the mid-pandemic. Hospitalizations decreased by 43% (95% CI 37.4–48.1) during the early pandemic and were still 26.6% (17.7–34.6) lower mid-pandemic. However, hospitalizations in non-psychiatric facilities for various mental health disorders increased substantially mid-pandemic.

### Conclusion

Healthcare utilization in children dropped substantially during the first year of the pandemic, with a shift away from infectious diseases and a spike in mental health hospitalizations.

University School of Medicine obtained this data for use by Stanford investigators. Because of this contractual relationship, Stanford investigators were allowed direct access to data. Investigators outside of Stanford interested in replicating this study should contact Optum directly for access. The Optum claims database website is https://www.optum.com/business/life-sciences/real-world-data/claims-data.html. Contact information can be found on the product sheet: https://cdn-aem.optum.com/content/dam/optum4/resources/pdf/clinformatics-data-mart.pdf.

**Funding:** ARS received funding for biostatistical support through a private fund (Michael Enright Fund). The donors of this fund played no role in study design, data collection and analysis, decision to publish, or preparation of the manuscript.

**Competing interests:** The authors have declared that no competing interests exist.

**Abbreviations:** US, United States; ED, Emergency Department; ICD, International Classification of Diagnosis; CPT, Current Procedural Terminology; PECCS, Pediatric Clinical Classification System; AHFS, American Hospital Formulary Service; ITS, Interrupted Time Series; POS: Place of Service; PCP, Primary Care Provider; ICU: CI, Confidence Interval; Intensive Care Unit; NPI, Non-pharmacologic Interventions; UTI, Urinary Tract Infection.

These findings are important to characterize as we monitor the health of children, can be used to inform healthcare strategies during subsequent COVID-19 surges and/or future pandemics, and may help identify training gaps for pediatric trainees. Subsequent investigations should examine how changes in healthcare utilization impacted the incidence and outcomes of specific diseases.

## Background

Early trends since the start of the COVID-19 pandemic suggested large-scale decreases in non-COVID-19 related healthcare utilization. These changes have been largely attributable to a decrease in transmission of common infections [1–8] and/or fear of COVID-19 exposure in the healthcare setting [9–13]. Though various investigations have described these changing patterns of utilization in children, most have analyzed specific conditions or settings of care (e.g. hospitalizations or prescription medication use), and few have assessed how patterns align across different types of utilization. Furthermore, the investigations into hospitalization trends have focused only on US children's hospitals, where fewer than half of all pediatric hospitalizations occur [14]. Additionally, the majority of these published studies address the time period immediately following the pandemic onset without evaluating ongoing effects as the pandemic evolved.

Insurance claims databases offer the opportunity to examine all types of utilization across the continuum of care for large populations of patients throughout the United States (US) [15–19]. Major shifts in health care use and diagnosis patterns as we have observed are important to quantify as we monitor child health following the pandemic. Additionally, quantification of these shifts help to identify potential gaps in clinical exposure for students, residents, and fellows who underwent training during the pandemic. Furthermore, characterization of utilization changes during this pandemic can inform healthcare strategies during future pandemics. Finally, trends in diagnoses across various utilization settings can identify the extent to which conditions are sensitive to changes brought about by the pandemic. Such information provides a "natural experiment" to investigate how viral transmission and healthcare utilization impact the incidence and outcomes of specific diseases, and quantifying any changes in incidence is a necessary first step. We aimed to use claims data to compare the rate of primary care, urgent care, and emergency department (ED) visits, hospitalizations, and outpatient prescription medications during the first year of the pandemic as compared to prior years. Additionally, we analyzed the pandemic's impact on common diagnoses and prescription medication use.

## Methods

### Design

Using a cross-sectional approach, we used Optum's De-identified Clinformatics Data Mart database to measure changes in utilization during vs before the pandemic. The Optum database contains administrative health claims for large commercial and Medicare Advantage health plans covering approximately 15–18 million lives annually, but does not include patients with Medicaid. The database includes patient demographics, pharmaceutical claims, inpatient and outpatient diagnosis codes, and procedure claims from enrollees in 50 states, with more than 2 million children enrolled each month [20]. The age and sex distribution of

the plan members is similar to that reported by the US Census Bureau [21]. Although deidentification of patient information makes formal validation of claims data challenging, some investigations have demonstrated encouraging reliability, though specificity of diagnostic codes is likely higher than sensitivity [22–24]. The Stanford University institutional review board approved the use of this deidentified database in this study under Stanford's Center for Population Health Sciences umbrella protocol. Given the deidentified nature of the dataset, informed consent was not required for this study.

## Time periods

We compared two time periods. The COVID-19 pandemic period covered mid-March, 2020 through March 31, 2021 while the pre-pandemic period included January 1, 2016 through mid-March, 2020 [25]. We elected to use 2016 as a starting point to allow sufficient time to account for secular trends in utilization that may have begun prior to the pandemic. For deidentification purposes, Optum only provides the month of service, not the specific date. Because most lockdowns began around mid-March 2020, we labeled March as being 50% in the pandemic period and 50% in the pre-pandemic period, and the pandemic period as starting in "mid-March".

We began the pre-pandemic period with the year 2016 because the International Classification of Diagnosis (ICD)-9 to ICD-10 conversion occurred in 2015, and because we surmised that 4 years is a sufficient time period for comparison. As utilization changes were dynamic during the pandemic, we further divided this period into two segments: "early pandemic" covered the first 3 months (mid-March 2020 –mid-June 2020) and "middle pandemic" covered mid-June 2020 through March 2021 (the most recently available data at the time of this investigation).

## Subjects

We included all subjects under 18 years of age who were enrolled in Optum during the study periods. Events were only included when occurring prior to the subject's 18[th] birthday. Census region and income quartile were determined by linking the patient's zip code to 2019 five-year American Community Survey US Census data.

## Utilization types

We compared primary care, urgent care, and ED visits, hospitalizations, and prescription medication claims. Our strategy for distinguishing visit types built upon prior investigations [17, 19, 26] and direct communication with the vendor. A combination of Current Procedural Terminology (CPT) and additional codes were used to distinguish utilization types as described in **S2 File**. Hospitalizations were characterized as psychiatric and non-psychiatric based on the designated provider category.

## Classifying diseases and medications

In order to categorize diagnoses, we grouped ICD-10 codes using the Pediatric Clinical Classification System (PECCS) classification system [27], and list the top 30 diagnoses by visit type in both the pre-pandemic and the pandemic time periods. We used the American Hospital Formulary Service (AHFS) classification system to categorize prescription medications and additionally categorize prescriptions as new or refill based on that designation in Optum. The top 30 prescription categories are assessed in each time period.

## Statistical analysis

For each visit and prescription type we calculated a rate by dividing the total number of visits or prescriptions in a month by the total number of patients (<18 years of age) who were covered by Optum during that month. We used an interrupted time series (ITS) approach to evaluate changes in healthcare utilization. Percent changes are measured with respect to a counterfactual where the pandemic had not occurred and the pre-pandemic seasonal fluctuations and linear trends had continued. Full details of the model specification are provided in (**S2 File**).

Finally, we also consider visit types associated with specific diseases using this same model. For ED and primary care provider (PCP) sick visits, and for hospitalizations, we report the changes in the number of visits associated with the most common PECCS disease classifications. For medication, we report the changes in the number of prescriptions written for the most common AHFS categories. For tables depicting the leading disease and prescription categories, we include any disease/medication that was in the top 30 during *either* the pre-pandemic period *or* the pandemic period. All analyses were conducted in Python, version 3.8.5. Regression parameters were estimated using the statsmodels package, version 0.12.0.

## Results

The mean number of child enrollees per month was 2,519,755 in the pre-pandemic period and 2,428,912 in the pandemic period. Demographic characteristics of the pediatric Optum cohort are provided in **Table 1**, which depicts comparisons of the pre-pandemic and pandemic periods. Overall the demographics varied minimally, with standardized mean differences less than 0.2. **S1 Table in S1 File** details results of an ITS analysis examining shifts in the demographic breakdown throughout the study period. Of note, the proportion of missing data on race in Optum has increased over recent years; however, the representation by category remained fairly constant for patients without missing data (**S1 Fig in S1 File**).

### Early pandemic utilization

During the early pandemic period, utilization decreased across all visit types and for hospitalizations and prescription fills (**Figs 1** and **2**). The largest drop occurred for urgent care visits (76.9% decrease, 95% confidence interval [CI] 72.6–80.5%), PCP sick visits (61.6% decrease, 95% CI 59.4% - 63.6%) and ED visits (59.6% decrease, 95% CI 54.8–63.9%) (**Fig 2**).

There were 288.9 prescription fills per 100 child years during the pre-pandemic period (with 54% of patients receiving at least one new or refill prescription in 2019). While prescription refills were unchanged during the early pandemic period, new prescriptions decreased by 47.1% (95% CI 43.7–50.3%).

Hospitalizations in non-psychiatric hospitals decreased by 43% (95% CI 37.4–48.1%), with similar decreases for hospitalizations that did and did not require an ICU stay. Hospitalizations in psychiatric facilities decreased by 36.6% (95% CI 31.9–40.9%).

### Middle pandemic utilization

During the mid-pandemic period, utilization patterns increased from the early pandemic period (**Fig 2**). However, with the exception of PCP well-visits, utilization during the middle pandemic period was still below the pre-pandemic rates, with the greatest sustained decreases seen in ED visits (38.4% decrease from pre-pandemic period, 95% CI 30.9–45.2%) and PCP sick visits (39% decrease, 95% CI 32.3–45.1%). After the initial decrease at the pandemic onset, new medication fills (**Fig 1**) recovered over the next 6 months, decreased again from

**Table 1. Demographic characteristics of the study population.**

| | Pre-Pandemic* | Pandemic* | Standardized Mean Difference** |
|---|---|---|---|
| **n** | 7,048,075 | 3,387,469 | |
| **Female** | 48.9% | 48.9% | 0.00 |
| **Age** | | | |
| < 1 | 2.2% | 2.1% | -0.04 |
| 1–5 | 25.9% | 25.6% | -0.01 |
| 6–12 | 40.6% | 40.4% | -0.01 |
| 13–17 | 31.3% | 31.9% | 0.02 |
| **Race/Ethnicity** (excluding unknown) | | | |
| non-Hispanic White | 69.7% | 70.5% | 0.04 |
| non-Hispanic Black | 8.1% | 7.9% | -0.04 |
| Hispanic | 15% | 14.4% | -0.05 |
| Asian | 7.2% | 7.2% | 0.01 |
| Unknown Race/Ethnicity | 16.5% | 42.3% | - |
| **Census Region** | | | |
| New England | 2.5% | 2.8% | 0.09 |
| Middle Atlantic | 6.4% | 6.4% | 0.00 |
| East North Central | 15.2% | 15.6% | 0.03 |
| West North Central | 11.4% | 13% | 0.15 |
| South Atlantic | 20% | 20.4% | 0.03 |
| East South Central | 3.7% | 3.8% | 0.03 |
| West South Central | 17.5% | 15.7% | -0.13 |
| Mountain | 10.9% | 10.8% | -0.01 |
| Pacific | 12.4% | 11.6% | -0.07 |
| **Income quartile** (of patient zip code) | | | |
| Lowest (< $55,019) | 21.8% | 21% | -0.04 |
| Second ($55,020 - $72,759) | 25.1% | 25.1% | 0.00 |
| Third ($72,759 - $96,969) | 26.7% | 27% | 0.02 |
| Highest (> $96,969) | 26.5% | 26.8% | 0.02 |

*Pre-pandemic period corresponds to January 2016 until mid-March 2020. Pandemic period corresponds to mid-March 2020 through March 2021.

**Standardized mean differences can be interpreted using Cohen's guideline: d = 0.2 corresponds to a small difference; d = 0.5, to a medium difference; and d = 0.8 to a large difference.

December 2020 –Feb 2021, and then started to increase again in March 2021. Over the entire middle pandemic period, new prescriptions were still 34.7% (95% CI 25.7–42.6%) lower than the pre-pandemic period.

Hospitalizations in non-psychiatric facilities returned closer to pre-pandemic rates during the middle-pandemic period but were still 26.6% (95% CI 17.7–34.6%) below baseline. Non-ICU hospitalizations recovered slightly more than hospitalizations with ICU stays, and hospitalizations in psychiatric facilities went nearly back to baseline (**Fig 2**).

In our sensitivity analysis, adjusting for the demographics portrayed in **Table 1** (age, sex, race, census region, and income quartile) did not have a meaningful impact on the changes in utilization in the early or mid-pandemic periods (**S6 Fig in S1 File**).

## Changes in common diagnoses

Common infectious diagnoses dropped substantially across utilization types during the early pandemic period and only increased modestly as the pandemic evolved. **Fig 3** displays heat

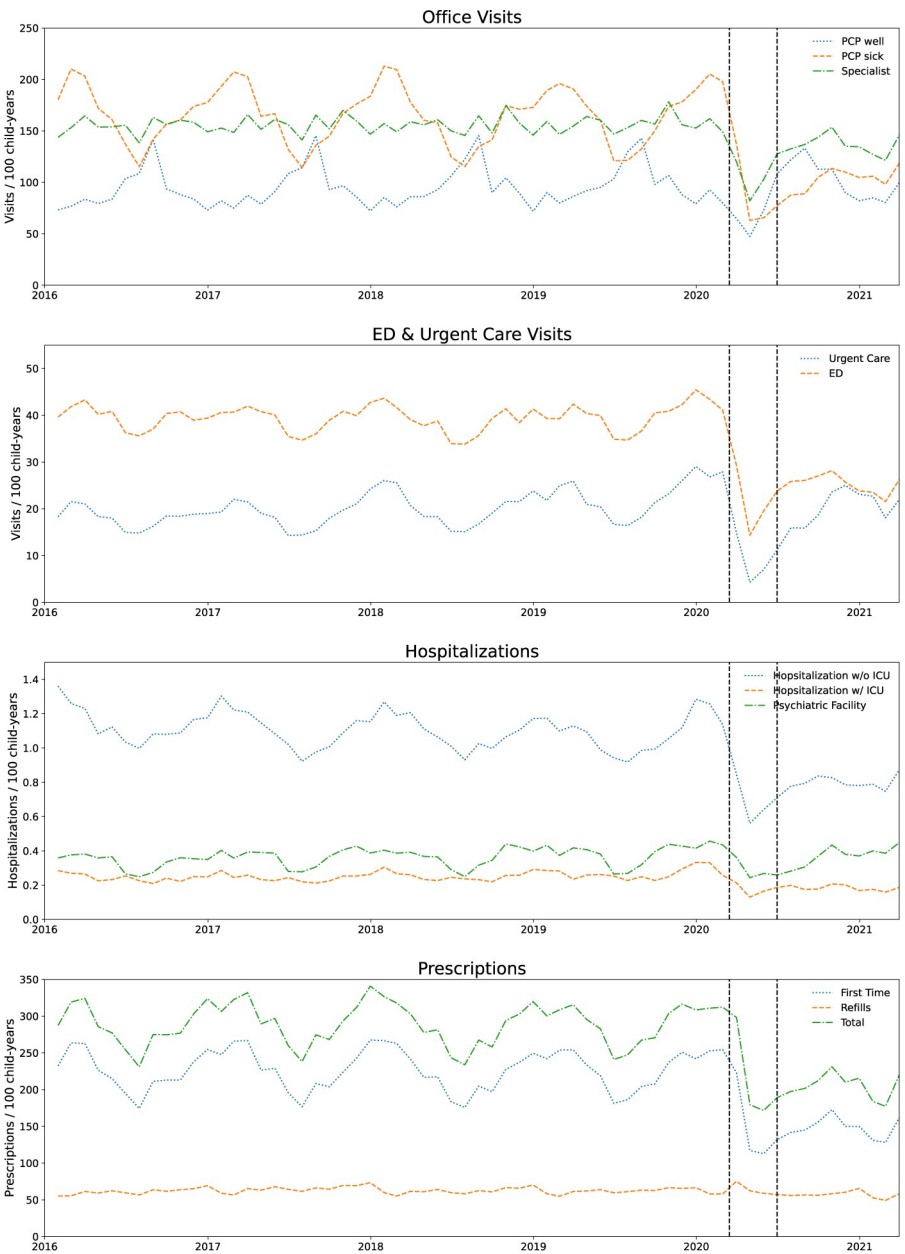

**Fig 1. Time series depicting trends in utilization and prescriptions per 100 child-years over the study period.**
Dotted vertical lines represent the starts of the early pandemic (mid-March) and mid-pandemic (mid-June).

maps representing the changes in the most common diagnoses associated with hospitalizations in non-psychiatric facilities. Notably, hospitalizations in non-psychiatric facilities for major depressive disorder (29.5% increase, 95% CI 16.6–43.8%), bipolar disorder (118.2% increase, 95% CI 40.5–239%), substance-related disorders (83.3% increase, 95% CI 6–217%) and suicide and intentional self-inflicted injury (82.1% increase, 95% CI 51.2–119.3%) were substantially increased from pre-pandemic levels during the middle pandemic period. Hospitalizations for some conditions that should not have been impacted by the reduction in transmissible viruses occurred less frequently during the pandemic. For example, neonatal hyperbilirubinemia

| Feature | Pre-pandemic count / 100 child-years | % Change during early pandemic* | % Change during mid-pandemic* |
|---|---|---|---|
| PCP well visit | 95.6 | -40.2% [-47.5% , -31.9%] | 1.6% [-4.7% , 8.3%] |
| PCP sick visit | 165.0 | -61.6% [-63.6% , -59.4%] | -39.0% [-45.1% , -32.3%] |
| ED | 39.4 | -59.6% [-63.9% , -54.8%] | -38.4% [-45.2% , -30.9%] |
| Urgent Care | 20.1 | -76.9% [-80.5% , -72.6%] | -24.8% [-32.2% , -16.5%] |
| Specialist | 156.1 | -44.2% [-48.4% , -39.7%] | -13.4% [-17.1% , -9.5%] |
| Hospitalizations | 1.3 | -43.0% [-48.1% , -37.4%] | -26.6% [-34.6% , -17.7%] |
| with ICU | 0.3 | -45.2% [-49.2% , -40.9%] | -34.8% [-43.1% , -25.2%] |
| without ICU | 1.1 | -42.4% [-48.1% , -36.2%] | -24.6% [-32.6% , -15.5%] |
| Admission to Psychiatric facilities | 0.4 | -36.6% [-40.9% , -31.9%] | -6.8% [-13.3% , 0.2%] |
| Prescriptions | 288.9 | -36.5% [-40.2% , -32.5%] | -28.7% [-35.9% , -20.7%] |
| first time | 226.2 | -47.1% [-50.3% , -43.7%] | -34.7% [-42.6% , -25.7%] |
| refills | 62.7 | 3.7% [-4.3% , 12.4%] | -7.3% [-12.9% , -1.4%] |

*Changes calculated using an interrupted time series approach
Early pandemic = mid-March 2020 – mid-June 2020
Mid-pandemic = mid-June 2020 through March 2021

**Fig 2. Pandemic changes in healthcare utilization with heat map.**

hospitalizations were 25.2% (95% CI 16.1–33.4%) lower during the early pandemic period and 31.8% (95% CI 23.1–39.5%) lower during the middle pandemic period.

**S2 Fig** in **S1 File** displays trends in common causes of hospitalizations in non-psychiatric facilities, PCP sick visits, ED visits, and prescriptions. **S3 Fig** in **S1 File** (PCP sick visits) and **S4 Fig** in **S1 File** (ED visits) display heat maps representing proportional changes in diagnoses

| Diagnosis | Rank Pre-Pandemic | Rank Post-Pandemic | % Change during early pandemic* | % Change during mid-pandemic* |
|---|---|---|---|---|
| Mood disorders (major depressive disorder) | 1 | 1 | -22.9% [-40.3% , -0.4%] | 29.5% [16.6% , 43.8%] |
| Acute bronchiolitis | 2 | 35 | N/A** | |
| Pneumonia | 3 | 16 | -88.7% [-92.8% , -82.3%] | -86.9% [-90.9% , -81.3%] |
| Chemotherapy | 4 | 2 | -3.5% [-13.6% , 7.9%] | -5.8% [-19.2% , 9.8%] |
| Asthma | 5 | 19 | -94.0% [-95.8% , -91.3%] | -69.6% [-77.9% , -58.2%] |
| Respiratory failure; insufficiency; arrest | 6 | 14 | -83.3% [-88.5% , -75.8%] | -82.2% [-88.5% , -72.2%] |
| Neonatal hyperbilirubinemia | 7 | 3 | -25.1% [-31.6% , -17.9%] | -33.2% [-40.0% , -25.6%] |
| Cellulitis | 8 | 12 | -25.2% [-37.6% , -10.3%] | -26.5% [-38.0% , -12.8%] |
| Acute appendicitis with peritonitis | 9 | 4 | -21.7% [-34.6% , -6.3%] | -16.3% [-34.5% , 6.8%] |
| Seizures w and w/o intractable epilepsy | 10 | 9 | -30.7% [-46.2% , -10.7%] | -16.2% [-37.7% , 12.6%] |
| Urinary tract infections | 11 | 8 | 2.0% [-14.6% , 21.9%] | -1.6% [-14.2% , 12.7%] |
| Mood disorders | 12 | 7 | 5.5% [-18.9% , 37.3%] | -3.2% [-24.6% , 24.3%] |
| Dehydration | 13 | 31 | -67.6% [-74.4% , -59.0%] | -64.8% [-74.3% , -51.8%] |
| Diabetic ketoacidosis | 14 | 6 | 6.3% [-9.9% , 25.4%] | 35.0% [9.6% , 66.3%] |
| Septicemia (except in labor) | 15 | 10 | -27.9% [-42.1% , -10.3%] | -16.3% [-41.0% , 18.7%] |
| Suicide and intentional self-inflicted injury | 16 | 5 | -10.6% [-23.7% , 4.8%] | 82.1% [51.2% , 119.3%] |
| Gastroenteritis, infectious | 17 | 27 | -56.3% [-69.6% , -37.0%] | -28.9% [-52.5% , 6.5%] |
| Scoliosis | 18 | 11 | -47.6% [-72.7% , 0.7%] | 33.9% [3.5% , 73.1%] |
| Partial epilepsy w with w/o intractable epilepsy | 19 | 13 | -25.9% [-43.7% , -2.5%] | 35.5% [-7.9% , 99.2%] |
| Viral infection | 20 | 38 | -64.1% [-72.9% , -52.4%] | -52.2% [-69.9% , -24.1%] |
| Mood disorders (bipolar disorder) | 21 | 18 | -27.8% [-51.7% , 8.0%] | 118.2% [40.5% , 239.0%] |
| Neutropenia | 22 | 24 | -42.4% [-51.0% , -32.4%] | -36.8% [-48.5% , -22.6%] |
| Fracture of lower limb | 23 | 15 | -22.0% [-39.0% , -0.3%] | -17.6% [-37.0% , 7.9%] |
| Specified conditions originating in perinatal … | 24 | 23 | -5.2% [-33.1% , 34.4%] | -22.0% [-49.6% , 20.7%] |
| Acute appendicitis w/o peritonitis | 25 | 28 | -36.7% [-53.1% , -14.7%] | -13.7% [-35.6% , 15.6%] |
| Complications of surgical procedures or medica… | 26 | 22 | -34.4% [-54.5% , -5.4%] | 16.5% [-23.8% , 78.3%] |
| Disturbances of temperature regulation of newb… | 27 | 32 | -55.2% [-64.8% , -43.1%] | -68.1% [-77.2% , -55.3%] |
| Headache; including migraine | 28 | 52 | -81.6% [-92.7% , -53.5%] | -35.3% [-52.8% , -11.2%] |
| Influenza | 29 | 355 | N/A** | |
| Respiratory distress syndrome in newborn | 30 | 21 | -7.1% [-29.2% , 22.0%] | 19.8% [-10.2% , 59.9%] |
| Infective arthritis and osteomyelitis (except caused by TB or STD) | 33 | 26 | 11.3% [-24.9% , 64.9%] | 5.4% [-37.9% , 78.9%] |
| Intracranial injury | 37 | 24 | -9.0% [-37.4% , 32.4%] | 7.5% [-35.6% , 79.5%] |
| Complications of device; implant or graft | 39 | 29 | -33.0% [-46.2% , -16.6%] | -9.8% [-48.7% , 58.6%] |
| Anorexia nervosa | 52 | 17 | 30.2% [-28.4% , 136.8%] | 98.4% [-13.8% , 356.8%] |
| Substance-related disorders | 57 | 30 | 35.8% [-17.1% , 122.3%] | 83.3% [6.0% , 217.0%] |
| Viral Infection (COVID-19) | - | 20 | N/A | |

*Changes calculated using an interrupted time series approach; Early pandemic = mid-March 2020 – mid-June 2020 and Mid-pandemic = mid-June 2020 through March 2021
** Model did not converge due to low numbers post-pandemic

**Fig 3. Pandemic changes in leading causes of hospitalization with heat map.**

| Diagnosis | Rank Pre-Pandemic | Rank Post-Pandemic | % Change during early pandemic* | % Change during mid-pandemic* |
|---|---|---|---|---|
| Aminopenicillins | 1 | 4 | -77.9% [-80.8% , -74.7%] | -66.9% [-74.1% , -57.7%] |
| Respiratory and CNS stimulants | 2 | 1 | -27.1% [-31.0% , -22.9%] | -13.7% [-18.3% , -8.9%] |
| Amphetamines | 3 | 2 | -25.1% [-28.5% , -21.6%] | -11.2% [-15.1% , -7.1%] |
| Selective beta-2-andrenergic agonists | 4 | 6 | -48.0% [-61.8% , -29.0%] | -55.9% [-65.4% , -43.7%] |
| Antibacterials (Eyes, Ears, Nose, Throat) | 5 | 8 | -65.8% [-69.3% , -61.9%] | -53.1% [-68.2% , -31.0%] |
| Other macrolides | 6 | 20 | -80.1% [-84.9% , -73.9%] | -74.1% [-78.0% , -69.6%] |
| Adrenals | 7 | 10 | -69.4% [-72.8% , -65.5%] | -58.7% [-65.8% , -50.2%] |
| Third generation cephalosporins | 8 | 19 | -78.8% [-82.5% , -74.2%] | -70.9% [-76.7% , -63.6%] |
| Selective-serotonin reuptake inhibitors | 9 | 3 | -4.3% [-10.0% , 1.8%] | 0.5% [-4.5% , 5.9%] |
| Anti-inflammatory agents (skin, mucous) | 10 | 5 | -16.6% [-20.0% , -13.0%] | 2.8% [-2.7% , 8.6%] |
| Antibacterials (skin & mucous membrane) | 11 | 7 | -23.7% [-26.6% , -20.8%] | -8.1% [-13.5% , -2.3%] |
| Leukotriene modifiers | 12 | 11 | -21.2% [-28.4% , -13.3%] | -31.2% [-39.0% , -22.3%] |
| Neuraminidase inhibitors | 13 | 83 | -98.2% [-99.1% , -96.5%] | -97.8% [-99.4% , -91.7%] |
| Orally inhaled preparations (steroids) | 14 | 12 | -7.3% [-24.0% , 13.1%] | -34.0% [-46.5% , -18.5%] |
| First generation cephalosporins | 15 | 16 | -37.1% [-40.4% , -33.7%] | -25.5% [-32.2% , -18.1%] |
| Serotonin receptor antagonists | 16 | 24 | -76.7% [-79.5% , -73.5%] | -60.8% [-67.4% , -52.9%] |
| Antitussives | 17 | 44 | -90.4% [-93.8% , -85.2%] | -84.2% [-87.5% , -80.0%] |
| Contraceptives | 18 | 9 | 0.3% [-5.0% , 5.9%] | -2.2% [-8.3% , 4.3%] |
| Corticosteroids (Eyes, Ears, Nose, Throat) | 19 | 26 | -45.1% [-50.9% , -38.6%] | -40.6% [-46.8% , -33.6%] |
| Opiate agonists | 20 | 25 | -34.5% [-53.1% , -8.4%] | 24.7% [6.3% , 46.2%] |
| Anticonvulsants, miscellaneous | 21 | 14 | -3.4% [-8.5% , 1.9%] | -2.7% [-6.9% , 1.7%] |
| Skin and mucous membrane agents, misc. | 22 | 18 | -18.7% [-25.3% , -11.5%] | -13.1% [-22.0% , -3.1%] |
| Sulfonamides (systemic) | 23 | 28 | -27.9% [-31.2% , -24.5%] | -16.5% [-20.0% , -12.9%] |
| Central nervous system agents, misc. | 24 | 15 | -9.6% [-14.9% , -4.0%] | -9.4% [-14.5% , -4.1%] |
| Atypical antipsychotics | 25 | 17 | -2.2% [-8.5% , 4.5%] | -1.3% [-6.0% , 3.6%] |
| Other nonsteroidal anti-inflam. agents | 26 | 27 | -53.2% [-60.7% , -44.4%] | -16.7% [-21.3% , -11.9%] |
| Tetracyclines | 27 | 22 | -13.0% [-17.5% , -8.4%] | 5.0% [-2.9% , 13.4%] |
| Devices | 28 | 29 | -28.1% [-35.2% , -20.1%] | -22.6% [-29.6% , -15.0%] |
| Central alpha-agonists | 29 | 21 | 2.6% [-1.6% , 7.1%] | 3.2% [-1.3% , 8.0%] |
| Alpha- and beta-adrenergic agonists | 30 | 30 | -47.5% [-55.5% , -37.9%] | -3.5% [-23.6% , 21.8%] |
| Anxiolytics, sedatives, and hypnotics, misc | 33 | 23 | -7.9% [-11.2% , -4.4%] | 4.2% [-0.7% , 9.3%] |
| Vaccines | 57 | 13 | -78.8% [-88.6% , -60.5%] | -60.8% [-82.8% , -10.6%] |

*Changes calculated using an interrupted time series approach; Early pandemic = mid-March 2020 – mid-June 2020 and
Mid-pandemic = mid-June 2020 through March 2021

**Fig 4. Pandemic changes in leading prescriptions with heat map.**

during the early and middle pandemic periods as compared to the pre-pandemic periods. Large decreases in infectious diagnoses were observed in both settings. However, in contrast to hospitalizations for mental health conditions, mental health diagnoses were generally not increased above pre-pandemic norms for ED or sick visits

## Changes in common new prescriptions

During the pre-pandemic period, antibiotics were the most commonly prescribed new medication, constituting 5 of the top 10 AHFS codes (**Fig 4**). During the pandemic, antibiotic prescriptions decreased substantially, and respiratory and central nervous system stimulants, amphetamines, selective serotonin reuptake inhibitors, and anxiolytics comprised 4 of the top 5 new prescription class codes. However, the vast majority of prescription medications were filled at rates below pre-pandemic baselines. Rates of selective-serotonin reuptake inhibitors (anti-depressants) were unchanged. The only medication that increased significantly was opiate agonists, which were 24.7% higher (95% CI 6.3–46.2%) during the middle-pandemic period.

## Telehealth

Prior to the pandemic, billed telehealth visits were rare (**S5 Fig in S1 File**). During the early pandemic, telehealth visits increased considerably, especially for PCP sick visits, where nearly half of these visits were conducted via telehealth. Telehealth visits decreased fairly quickly as the pandemic evolved, though remained more frequent than the pre-pandemic period.

## Discussion

In this analysis of over 2 million children enrolled per year in a US insurance claims database, we report striking changes in healthcare utilization by children during the COVID-19 pandemic. Utilization decreased across all sites of care but was more pronounced for sick visits (at PCPs, urgent cares, and EDs) than for well-child visits. Decreases persisted throughout the first 12 months of the pandemic, with only well-child visits returning to their pre-pandemic rates. Common diagnoses associated with these healthcare encounters shifted away from infectious conditions, with lesser decreases or even increases (hospitalizations in non-psychiatric facilities) in mental health conditions. Similarly, prescription fills decreased with especially large drops in antibiotics.

Drivers of the observed patterns are likely multifactorial. Physical distancing and other non-pharmacologic interventions (NPIs) to prevent the spread of SARS-CoV-2 led to a decrease in circulating infectious organisms that are common causes of sick visits, hospitalizations, and antibiotic use in children [1–8]. Additionally, families may have been more reluctant to pursue healthcare for fear of exposure to SARS-CoV-2. Lastly, the stresses imposed by the pandemic and measures to mitigate its severity likely triggered the observed increases in mental health hospitalizations in non-psychiatric facilities. However, hospitalizations in psychiatric hospitals did not increase over the first year, perhaps because psychiatric hospitals often operate at or near maximal occupancy [28].

There have been multiple related investigations describing pandemic associated patterns of utilization. However, the majority focus on unique utilization types and have shorter pandemic analysis periods. Using claims data, Schweiberger et al. [19] demonstrated similar initial drops in utilization, with a more pronounced decrease in infectious-related visits. Similar decreases have been demonstrated consistently in studies of ER visits [29–32].

Multiple investigations have demonstrated a reduction in hospitalizations [2, 3, 33–36], but most are limited to children's hospitals despite the fact that the majority of pediatric hospitalizations in the US occur outside of children's hospitals [14]. Our alternative approach may explain why we found different trends for certain conditions. For example, we found that urinary tract infection (UTI) hospitalizations remained fairly stable, whereas Gill et al. [3] demonstrated a reduction of UTI hospitalizations in children's hospitals. Whether this discrepancy might be explained by inclusion of non-children's hospitals in our investigation, inclusion of public insurance in their investigation, or other factors warrants further investigation.

For prescription fills, Chua et al. [37] analyzed a national prescription audit of US pharmacies and demonstrated a similar pattern of a dramatic drop in prescriptions early on in the pandemic, a gradual rise over the next several months, but then a drop again in December. We demonstrate that this drop (which may have been attributable to heightened NPI efforts during that December surge) continued through January and February of 2021 but then increased again. Similar to that investigation, we observed large decreases in antibiotics and no significant change in anti-depressant use (though mental health medications in general comprised a higher proportion of the most common prescriptions). The fact that the rise in hospitalizations for mental health disorders was not accompanied by an increase (in absolute terms) in prescriptions for depression and other mental health disorders is noteworthy and warrants further investigation. We did observe a significant increase in opiate prescriptions during the mid-pandemic period, along with a significant increase in hospitalizations for substance-related disorders, concerning findings that warrant further investigation and continued monitoring.

This comprehensive assessment of healthcare utilization in children during the pandemic can inform the literature in several ways. First, in the event of future lockdowns for this or other pandemics, our data can be used to assist with predictions surrounding pediatric

healthcare needs. Given the persistence of SARS-CoV-2, the emergence of new variants, and the possibility of future pandemics caused by other organisms, our findings may be useful to anticipate where and how scarce resources are most needed. For example, early appropriation of resources towards mental health services may help to mitigate this important morbidity. Additionally, the near-disappearance of circulating viruses and gradual return may inform immunization strategies (e.g. for influenza or respiratory syncytial virus immunoprophylaxis).

Second, this work provides the foundation to further explore the necessity of care for various conditions [38]. For diseases that may have been impacted by a decreased pursuit of healthcare, changes in diagnostic patterns may provide further insight into the natural history of untreated disease and the extent to which the timing of diagnosis impacts outcomes. For example, delays in the pursuit of healthcare may have contributed to the increase in hospitalizations for diabetic ketoacidosis (**Fig 3**), which highlights the need for increased vigilance around this condition. However, for other conditions, the decreased pursuit of healthcare may not have had adverse consequences. Notably, hyperbilirubinemia hospitalizations decreased considerably during the pandemic (**Fig 3**). Because lockdowns and NPIs (and the resulting decrease in transmission of circulating viruses) should have had minimal if any impact on the true prevalence of neonatal hyperbilirubinemia, this reduction in hospitalizations may be explained by heightened efforts to keep young infants out of the hospital. Whether this reduction was associated with an increase in severe outcomes warrants further evaluation that may ultimately inform strategies for treatment of jaundice. Similarly, we noted that UTI diagnoses decreased substantially in the ED setting (**S4 Fig in S1 File**), though hospitalization rates remained similar. Because UTIs are not thought to be transmissible, these changes are also worth exploring to better understand the epidemiology of UTI–a condition which may self-resolve without treatment [39] and where uncertainty over the gold standard for diagnosis may contribute to high rates of false positives [40].

Third, our data can be used to inform our understanding of certain diseases in children where the etiology remains uncertain and where viruses may play a role. Examples of such conditions include Type I diabetes [41], Henoch-Schoenlein Purpura [42], appendicitis [43], and certain malignancies [44]. Because the overall incidence of viral infections decreased so dramatically during the pandemic, especially early on, conditions that are acutely or sub-acutely triggered by viruses should have decreased as well. We noted, for example, that appendicitis (with or without peritonitis) hospitalizations decreased (**Fig 3**). Possible explanations for this finding include a potential role of viruses in triggering appendicitis or the possibility that milder cases of appendicitis may have gone undetected and self-resolved. Although the current investigation will not provide definitive answers to these questions, we hope that our data will be used to stimulate hypotheses and further investigations.

Fourth, we provide general descriptive information on overall healthcare use in children before and during the pandemic. This information can be used to guide and target quality improvement and de implementation efforts. We found that privately insured US children, on average, have 3 prescriptions filled per year, and that just over half of children have at least one prescription filled per year is quite striking. In comparison, only ~ 23% of Canadian children had at least one prescription fill per year between 2012–2017 [45]. Whether this contrast reflects differences in health, health care practices, or other factors warrants further investigation.

This investigation has several limitations. Most importantly, the claims database only includes children with commercial insurance and utilization patterns may differ for publicly insured or uninsured children. Although there are active efforts to procure Medicaid claims database at Stanford, the investigators did not have access to public insurance databases at the time of this investigation. Understanding how these utilization patterns differed across

demographics is an important next step. Similarly, investigations comparing shifts in utilization patterns between different countries and in the context of varying states of transmission of COVID-19 (and related restrictions) may also help to inform policy.

We relied on ICD-10 codes for diagnoses. Although investigations into the validity of diagnostic codes in claims data have mostly been promising [22–24], there is still potential for miscoding. Additionally, as our intent for this investigation was to provide a broad overview of utilization and diagnostic shifts, we only included the first diagnosis in each claim. Inclusion of all diagnoses should be considered to comprehensively investigate changes for specific diseases. Furthermore, we did not attempt to discriminate between new versus recurrent problems. Such discrimination is important to better understand some of the changes we observed in conditions such as diabetic ketoacidosis. Lastly, our division of the first year of the pandemic into two periods was based on our initial visualization of the data. Separation of the pandemic into different periods may have yielded different point estimates for utilization changes and may have impacted the list of top diagnoses. For utilization changes, the monthly rates depicted in **Fig 1** and **S2 Fig in S1 File** provide the most detailed and accurate picture of utilization trends.

## Conclusion

The COVID-19 pandemic has been associated with substantial decreases changes in healthcare utilization in children. These changes have important implications for healthcare policy and disease epidemiology.

## Supporting information

**S1 File. Additional supporting information is provided in the supplement, which includes S1 Table and S1-S6 Figs.**
(PDF)

**S2 File. Supplemental methods PUPPI resubmission clean.**
(DOCX)

## Author Contributions

**Conceptualization:** Alan R. Schroeder, Lauren Destino, Eric R. Coon.

**Data curation:** Natasha Purington.

**Formal analysis:** Alan R. Schroeder, Alex Dahlen, Natasha Purington, Eric R. Coon.

**Funding acquisition:** Alan R. Schroeder.

**Investigation:** Alan R. Schroeder.

**Methodology:** Alan R. Schroeder, Alex Dahlen, Natasha Purington, Francisco Alvarez, Rona Brooks, Lauren Destino, Gayatri Madduri, Marie Wang, Eric R. Coon.

**Writing – original draft:** Alan R. Schroeder.

**Writing – review & editing:** Alex Dahlen, Natasha Purington, Francisco Alvarez, Rona Brooks, Lauren Destino, Gayatri Madduri, Marie Wang, Eric R. Coon.

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
