## [Decision Letter · Decision Letter 0]

8 Jul 2022

PONE-D-22-14626Healthcare Utilization in Children Across the Care Continuum During the COVID-19 PandemicPLOS ONE

Dear Dr. Schroeder,

Thank you for submitting your manuscript to PLOS ONE. After careful consideration, we feel that it has merit but does not fully meet PLOS ONE’s publication criteria as it currently stands. Therefore, we invite you to submit a revised version of the manuscript that addresses the points raised during the review process. Please submit your revised manuscript by Aug 22 2022 11:59PM. If you will need more time than this to complete your revisions, please reply to this message or contact the journal office at plosone@plos.org. Please include the following items when submitting your revised manuscript:A rebuttal letter that responds to each point raised by the academic editor and reviewer(s). You should upload this letter as a separate file labeled 'Response to Reviewers'.A marked-up copy of your manuscript that highlights changes made to the original version. You should upload this as a separate file labeled 'Revised Manuscript with Track Changes'.An unmarked version of your revised paper without tracked changes. You should upload this as a separate file labeled 'Manuscript'.If applicable, we recommend that you deposit your laboratory protocols in protocols.io to enhance the reproducibility of your results. Protocols.io assigns your protocol its own identifier (DOI) so that it can be cited independently in the future. For instructions see: https://journals.plos.org/plosone/s/submission-guidelines#loc-laboratory-protocols. Additionally, PLOS ONE offers an option for publishing peer-reviewed Lab Protocol articles, which describe protocols hosted on protocols.io. Read more information on sharing protocols at https://plos.org/protocols?utm_medium=editorial-email&utm_source=authorletters&utm_campaign=protocols.

We look forward to receiving your revised manuscript.

Kind regards,

Tai-Heng Chen, M.D.

Academic Editor

PLOS ONE

Journal Requirements:

"NO authors have competing interests"

Reviewers' comments:

Reviewer's Responses to Questions

**Comments to the Author**

1. Is the manuscript technically sound, and do the data support the conclusions?

Reviewer #1: Yes

Reviewer #2: Yes

2. Has the statistical analysis been performed appropriately and rigorously? 

Reviewer #1: Yes

Reviewer #2: Yes

3. Have the authors made all data underlying the findings in their manuscript fully available?

Reviewer #1: Yes

Reviewer #2: No

4. Is the manuscript presented in an intelligible fashion and written in standard English?

Reviewer #1: Yes

Reviewer #2: Yes

5. Review Comments to the Author

Reviewer #1: Healthcare Utilization in Children Across the Care Continuum During the COVID-19 Pandemic

Overall:

The paper focused on the essential topic of “ utilization of pediatric services during Covid”. It is week written.

Introduction:

The information included in the second paragraph does not have any reference. In the second paragraph, “Major shifts in health care use and diagnosis patterns are important to describe as we monitor child health following the pandemic” needs to be rephrased. What shift the authors are referring to is unclear. How the training can help the shift has also not been clarified. The statement “how viral transmission and healthcare utilization impact the incidence and outcomes of specific diseases” is beyond the scope of this study as the source of data is insurance claim administrative data.

Method

The major concern is why include four years of data for comparison rather than taking previous year's data (March 2019-Feb 2020) for analysis? There was no justification given. The authors focused on primary care, urgent care, and ED visits, hospitalizations, and prescription medication claim but did not provide the operational definition of those terms.

Results

It would have been useful for the policymakers to show the utilization rate change according to the income quintile, race, and gender.

Discussion

The utilization rate over four years period might create a bias toward utilization rate. The conclusion about future policy or strategy the differentiation if any, across gender, race, and income quintile would have been helpful.

Reviewer #2: This is a very well-written manuscript describing changes in healthcare utilization related to the pandemic across different sectors in a population-based sample of US children. It adds to the literature by broadening the US data to include outpatient care and prescriptions and by including emergency department and admissions to community hospitals. Given the broad nature of the outcomes, the analyses raise more questions that are relevant both for future pandemics but also to potentially extraneous patterns of health care use at baseline. The authors did an excellent job in the discussion of highlighting which findings require more investigation.

I have only minor comments to potentially strengthen the manuscript.

Methods:

• As per RECORD guidelines, a comment on whether the dataset has ever been validated would be helpful (even if it hasn’t been – there was only a comment in the limitations in general related to ICD-10 codes).

Analysis:

• it would be important for the authors to justify why the models weren’t age/sex adjusted – the fact that age and sex distribution were similar amongst enrollees across the time periods doesn’t mean that there would not have potentially been effects of either across the outcomes.

Discussion:

• The authors cite studies from other jurisdictions (UK, Canada etc) and it would be helpful to comment that the consistencies of many findings, despite different 1) health systems, 2) levels of COVID transmission at various time points and 3) restrictions (esp school closures) is also worthy of more comparative investigation.

• The inability to distinguish new vs recurrent problems (because of no ability to link across episodes) is a limitation that should be included. This is esp. true for the findings around DKA as the implications of higher rates of DKA in incident vs prevalent cases are different. For the former this is related to delays in care seeking or issues in misdiagnoses in primary care whereas for children with known diabetes, this would be related to problems with specialized care.

6. PLOS authors have the option to publish the peer review history of their article (what does this mean?). If published, this will include your full peer review and any attached files.

Reviewer #1: No

Reviewer #2: **Yes: **Astrid Guttmann

---

## [Author Response · Author response to Decision Letter 0]

23 Aug 2022

Dear Editors, PLoS One:

We appreciate the favorable reviews for our manuscript and thank the reviewers for their thoughtful comments and suggestions, which have strengthened the manuscript. Below is a point-by-point response to each suggestion. 

Reviewer #1: 

Comment 1: The information included in the second paragraph does not have any reference.

Response: Multiple additional references were added following the first sentence. 

Comment 2: In the second paragraph, “Major shifts in health care use and diagnosis patterns are important to describe as we monitor child health following the pandemic” needs to be rephrased. What shift the authors are referring to is unclear.

Response: The sentence has been modified and now reads as follows: “Major shifts in health care use and diagnosis patterns as we have observed are important to describe quantify as we monitor child health following the pandemic.”

Comment 3: How the training can help the shift has also not been clarified. 

Response: The sentence has been modified and now reads: “Additionally, quantification of these shifts help to identify potential gaps in clinical exposure for students, residents, and fellows who underwent training during the pandemic.”

Comment 4: The statement “how viral transmission and healthcare utilization impact the incidence and outcomes of specific diseases” is beyond the scope of this study as the source of data is insurance claim administrative data.

Response: We agree that we are not analyzing outcomes in this paper. However, this paper will serve as a guide for our group (and hopefully other investigators) to know which conditions to target for these types of investigations. We have modified the sentence as follows: “Such information provides a “natural experiment” to investigate how viral transmission and healthcare utilization impact the incidence and outcomes of specific diseases, and quantifying any changes in incidence is a necessary first step.”

Comment 5: The major concern is why include four years of data for comparison rather than taking previous year's data (March 2019-Feb 2020) for analysis? There was no justification given. 

Response: Thank you for this comment – we agree that clarification would help. We felt that we needed at least a few years of pre-pandemic data to account for trends that may have predated the pandemic (for example, SSRI prescriptions were rising steadily from 2016-2019) and to mitigate the impact that an unusual occurrence (e.g. a bad influenza year) would have if only one year were used. We did not go back further than 2016 given the ICD9 to ICD10 transition in late 2015. We have added the following sentence: “We elected to use 2016 as a starting point to allow sufficient time to account for secular trends in utilization that may have begun prior to the pandemic.”

Comment 6: The authors focused on primary care, urgent care, and ED visits, hospitalizations, and prescription medication claim but did not provide the operational definition of those terms.

Response: The first page of Supplementary Information describes how all of these types of utilization were defined. 

Comment 7: It would have been useful for the policymakers to show the utilization rate change according to the income quintile, race, and gender.

Response: We agree with the reviewer that our analysis leaves open the possibility that different demographic groups were affected differently during the pandemic—heterogeneity of treatment effect. While a full analysis of this heterogeneity would be really interesting, we decided that it was outside the scope of our analysis, and would require multiple additional tables and figures, as well as potentially lengthy explanations for any notable findings. We are in the process of procuring a Medicaid database and do intend to compare and contrast utilization patterns for that. 

Nonetheless, given the related comment from Reviewer 2 below, we did do an additional analysis adjusting for these covariates. See comments below.

We have added a sentence to limitations section after mentioning the private insurance only limitation: “Understanding how these utilization patterns differed across demographics is an important next step.”

Comment 8: The utilization rate over four years period might create a bias toward utilization rate. 

Response: The utilization rate described is a count per 100 child years, and should not be biased by the number of years prior to the pandemic. As stated above, it is important to account for trends that predated the pandemic, and the ITS model is used precisely for that purpose. 

Comment 9: The conclusion about future policy or strategy the differentiation if any, across gender, race, and income quintile would have been helpful.

Response: thank you for this insightful comment. Please see above responses and response to reviewer comment 2. 

Reviewer #2: 

We thank Dr. Guttman for her very kind comments and suggestions for improvement. 

Comment 1: As per RECORD guidelines, a comment on whether the dataset has ever been validated would be helpful (even if it hasn’t been – there was only a comment in the limitations in general related to ICD-10 codes).

Response: We have added a sentence to Methods, with 3 references investigating this question: “Although deidentification of patient information makes formal validation of claims data challenging, some investigations have demonstrated encouraging reliability, though specificity of diagnostic codes is likely higher than sensitivity.22-24”

The section in Limitations reads: “Although investigations into the validity of diagnostic codes in claims data have mostly been promising,22-24 there is still potential for miscoding.”

Comment 2: it would be important for the authors to justify why the models weren’t age/sex adjusted – the fact that age and sex distribution were similar amongst enrollees across the time periods doesn’t mean that there would not have potentially been effects of either across the outcomes.

Response: A similar comment was made by Reviewer 1, see our related response above. We focused on unadjusted differences in part because presentation in this fashion can be easier to interpret. Additionally, we wanted our data to depict actual “real world” changes. When we performed the full interrupted time series analysis for trends in demographic makeup cohort (results in the supplement), we found that no demographic variable made a step change of >1%. Nonetheless, in response to this suggestion, we’ve also added a sensitivity analysis to the Supplement where we adjust for demographic shifts, and indeed the results are pretty much the same.

The Supplementary Methods now ends with: “Lastly, as a sensitivity analysis, we included demographic covariates (age, sex, and race, census region, and income quartile) in the ITS model to examine the extent to which adjustment for these variables led affected the unadjusted rates.”

The Results section now has a new paragraph reading: “In our sensitivity analysis, adjusting for the demographics portrayed in Table 1 (age, sex, race, census region, and income quartile) did not have a meaningful impact on the changes in utilization in the early or mid-pandemic periods (Supplementary Figure 6).” 

The supplementary tables and figures now has an additional Supplementary Figure 6 portraying the results of this sensitivity analysis. 

Comment 3: The authors cite studies from other jurisdictions (UK, Canada etc) and it would be helpful to comment that the consistencies of many findings, despite different 1) health systems, 2) levels of COVID transmission at various time points and 3) restrictions (esp school closures) is also worthy of more comparative investigation.

Response: We added the following sentence to Limitations: “Similarly, investigations comparing shifts in utilization patterns between different countries and in the context of varying states of transmission of COVID-19 (and related restrictions) may also help to inform policy.”

Comment 4: The inability to distinguish new vs recurrent problems (because of no ability to link across episodes) is a limitation that should be included. This is esp. true for the findings around DKA as the implications of higher rates of DKA in incident vs prevalent cases are different. For the former this is related to delays in care seeking or issues in misdiagnoses in primary care whereas for children with known diabetes, this would be related to problems with specialized care.

Response: We added the following two sentences to Limitations: “Furthermore, we did not attempt to discriminate between new versus recurrent problems. Such discrimination is important to better understand some of the changes we observed in conditions such as diabetic ketoacidosis.”

---

## [Decision Letter · Decision Letter 1]

7 Oct 2022

Healthcare Utilization in Children Across the Care Continuum During the COVID-19 Pandemic

PONE-D-22-14626R1

Dear Dr. Schroeder,

We’re pleased to inform you that your manuscript has been judged scientifically suitable for publication and will be formally accepted for publication once it meets all outstanding technical requirements.

Kind regards,

Tai-Heng Chen, M.D.

Academic Editor

PLOS ONE

Reviewers' comments:

Reviewer's Responses to Questions

**Comments to the Author**

1. If the authors have adequately addressed your comments raised in a previous round of review and you feel that this manuscript is now acceptable for publication, you may indicate that here to bypass the “Comments to the Author” section, enter your conflict of interest statement in the “Confidential to Editor” section, and submit your "Accept" recommendation.

Reviewer #1: All comments have been addressed

2. Is the manuscript technically sound, and do the data support the conclusions?

Reviewer #1: Yes

3. Has the statistical analysis been performed appropriately and rigorously? 

Reviewer #1: Yes

4. Have the authors made all data underlying the findings in their manuscript fully available?

Reviewer #1: No

5. Is the manuscript presented in an intelligible fashion and written in standard English?

Reviewer #1: Yes

6. Review Comments to the Author

Reviewer #1: All comments are addressed. There are several limitations of this study which have been acknowledged by the authors!!

7. PLOS authors have the option to publish the peer review history of their article (what does this mean?). If published, this will include your full peer review and any attached files.

Reviewer #1: **Yes: **Malabika Sarker

---

## [Editor Report · Acceptance letter]

20 Oct 2022

PONE-D-22-14626R1 

Healthcare Utilization in Children Across the Care Continuum During the COVID-19 Pandemic 

Dear Dr. Schroeder:

I'm pleased to inform you that your manuscript has been deemed suitable for publication in PLOS ONE. Congratulations! Your manuscript is now with our production department. 

Kind regards, 

on behalf of

Dr. Tai-Heng Chen 

Academic Editor

PLOS ONE